# PDE-NET: LEARNING PDEs FROM DATA

## ABSTRACT

Partial differential equations (PDEs) play a prominent role in many disciplines such as applied mathematics, physics, chemistry, material science, computer science, etc. PDEs are commonly derived based on physical laws or empirical observations. However, the governing equations for many complex systems in modern applications are still not fully known. With the rapid development of sensors, computational power, and data storage in the past decade, huge quantities of data can be easily collected and efficiently stored. Such vast quantity of data offers new opportunities for data-driven discovery of hidden physical laws. Inspired by the latest development of neural network designs in deep learning, we propose a new feed-forward deep network, called PDE-Net, to fulfill two objectives at the same time: to accurately predict dynamics of complex systems and to uncover the underlying hidden PDE models. The basic idea of the proposed PDE-Net is to learn differential operators by learning convolution kernels (filters), and apply neural networks or other machine learning methods to approximate the unknown nonlinear responses. Comparing with existing approaches, which either assume the form of the nonlinear response is known or fix certain finite difference approximations of differential operators, our approach has the most flexibility by learning both differential operators and the nonlinear responses. A special feature of the proposed PDE-Net is that all filters are properly constrained, which enables us to easily identify the governing PDE models while still maintaining the expressive and predictive power of the network. These constrains are carefully designed by fully exploiting the relation between the orders of differential operators and the orders of sum rules of filters (an important concept originated from wavelet theory). We also discuss relations of the PDE-Net with some existing networks in computer vision such as Network-In-Network (NIN) and Residual Neural Network (ResNet). Numerical experiments show that the PDE-Net has the potential to uncover the hidden PDE of the observed dynamics, and predict the dynamical behavior for a relatively long time, even in a noisy environment.

## 1    INTRODUCTION

Differential equations, especially partial differential equations(PDEs), play a prominent role in many disciplines to describe the governing physical laws underlying a given system of interest. Traditionally, PDEs are derived based on simple physical principles such as conservation laws, minimum energy principles, or based on empirical observations. Important examples include the Navier-Stokes equations in fluid dynamics, the Maxwell's equations for electromagnetic propagation, and the Schrödinger's equations in quantum mechanics. However, many complex systems in modern applications (such as many problems in climate science, neuroscience, finance, etc.) still have eluded mechanisms, and the governing equations of these systems are only partially known. With the rapid development of sensors, computational power, and data storage in the last decade, huge quantities of data can be easily collected and efficiently stored . Such vast quantity of data offers new opportunities for data-driven discovery of potentially new physical laws. Then, one may ask the following interesting and intriguing question: can we learn a PDE model (if there exists one) from a given data set and perform accurate and efficient predictions using the learned model?

One of earlier attempts on data-driven discovery of hidden physical laws is by Bongard & Lipson (2007) and Schmidt & Lipson (2009). Their main idea is to compare numerical differentiations of the experimental data with analytic derivatives of candidate functions, and apply the symbolic regression and the evolutionary algorithm to determining the nonlinear dynamical system. Recently,

Brunton et al. (2016), Schaeffer (2017), Rudy et al. (2017) and Wu & Zhang (2017) propose an alternative approach using sparse regression. They construct a dictionary of simple functions and partial derivatives that are likely to appear in the unknown governing equations. Then, they take advantage of sparsity promoting techniques to select candidates that most accurately represent the data. When the form of the nonlinear response of a PDE is known, except for some scalar parameters, Raissi & Karniadakis (2017) presented a framework to learn these unknown parameters by introducing regularity between two consecutive time step using Gaussian process. More recently, Raissi et al. (2017) introduced a new class of universal function approximators called the physics informed neural networks which is capable of discovering nonlinear PDEs parameterized by scalars.

These recent work greatly advanced the progress of the problem. However, symbolic regression is expensive and does not scale very well to large systems. The sparse regression method requires to fix certain numerical approximations of the spatial differentiations in the dictionary beforehand, which limits the expressive and predictive power of the dictionary. Although the framework presented by Raissi & Karniadakis (2017); Raissi et al. (2017) is able to learn hidden physical laws using less data than the approach of sparse regression, the explicit form of the PDEs are assumed to be known except for a few scalar learnable parameters. Therefore, extracting governing equations from data in a less restrictive setting remains a great challenge.

The main objective of this paper is to accurately predict the dynamics of complex systems and to uncover the underlying hidden PDE models (should they exist) at the same time, with minimal prior knowledge on the systems. Our inspiration comes from the latest development of deep learning techniques in computer vision. An interesting fact is that some popular networks in computer vision, such as ResNet(He et al., 2016a;b), have close relationship with PDEs (Chen et al., 2015; E, 2017; Haber & Ruthotto, 2017; Sonoda & Murata, 2017; Lu et al., 2017). Furthermore, the deeper is the network, the more expressive power the network possesses, which may enable us to learn more complex dynamics arose from fields other than computer vision. However, existing deep networks designed in deep learning mostly emphasis on expressive power and prediction accuracy. These networks are not transparent enough to be able to reveal the underlying PDE models, although they may perfectly fit the observed data and perform accurate predictions. Therefore, we need to carefully design the network by combining knowledge from deep learning and applied mathematics so that we can learn the governing PDEs of the dynamics and make accurate predictions at the same time. Note that our work is closely related to Chen et al. (2015) where the authors designed their network based on discretization of quasilinear parabolic equations. However, it is not clear if the dynamics of image denoising has to be governed by PDEs, nor did the authors attempt to recover the PDE (should there exists one).

In this paper, we design a deep feed-forward network, named PDE-Net, based on the following generic nonlinear evolution PDE

$$u_t = F(x, u, \nabla u, \nabla^2 u, \ldots), \quad x \in \Omega \subset \mathbb{R}^2, \quad t \in [0, T].$$

The objective of the PDE-Net is to learn the form of the nonlinear response $F$ and to perform accurate predictions. Unlike the existing work, the proposed network only requires minor knowledge on the form of the nonlinear response function $F$, and requires no knowledge on the involved differential operators (except for their maximum possible order) and their associated discrete approximations. The nonlinear response function $F$ can be learned using neural networks or other machine learning methods, while discrete approximations of the differential operators are learned using convolution kernels (i.e. filters) jointly with the learning of the response function $F$. If we have a prior knowledge on the form of the response function $F$, we can easily adjust the network architecture by taking advantage of the additional information. This may simplify the training and improve the results. We will also discuss relations of the PDE-Net to some existing networks in computer vision such as Network-In-Network (NIN) and ResNet. Details are given in Section 2.

In Section 3 and Section 4, we conduct numerical experiments on a linear PDE (convection-diffusion equation) and a nonlinear PDE (convection-diffusion equation with a nonlinear source). We generate data set for each PDE using high precision numerical methods and add Gaussian noise to mimic real situations. Our numerical results show that the PDE-Net can uncover the hidden equations of the observed dynamics, and can predict the dynamical behavior for a relatively long time, even in a noisy environment.

A particular novelty of our approach is that we impose appropriate constraints on the learnable filters in order to easily identify the governing PDE models while still maintaining the expressive and pre-

dictive power of the network. This makes our approach different from existing deep convolutional networks which mostly emphasis on the prediction accuracy of the networks, as well as all the existing approaches of learning PDEs from data which assume either the form of the response function is known or have fixed approximations of the differential operators. In other words, our proposed approach not only has vast flexibility in fitting observed dynamics and is able to accurately predict its future behavior, but is also able to reveal the hidden equations driving the observed dynamics. The constraints on the filters are motivated by the earlier work of Cai et al. (2012); Dong et al. (2017) where general relations between wavelet frame transforms and differential operators were established. In particular, it was observed in Dong et al. (2017) that we can relate filters and finite difference approximation of differential operators by examining the orders of sum rules of the filters (an important concept in wavelet theory and closely related to vanishing moments of wavelet functions). These constraints on the filters may also be useful in network designs for machine learning tasks in computer vision.

## 2 PDE-NET: A FLEXIBLE DEEP ARCHTECTURE TO LEARN PDEs FROM DATA

Given a series of measurements of some physical quantities $\{u(t, \cdot) : t = t_0, t_1, \cdots\}$ on the spatial domain $\Omega \subset \mathbb{R}^2$, with $u(t, \cdot) : \Omega \mapsto \mathbb{R}$, we want to discover the governing PDEs of the data. We assume that the observed data are associated with a PDE that takes the following general form:

$$u_t(t, x, y) = F(x, y, u, u_x, u_y, u_{xx}, u_{xy}, u_{yy}, \ldots), \quad (x, y) \in \Omega \subset \mathbb{R}^2, t \in [0, T]. \quad (1)$$

Our objective is to design a feed-forward network, named the PDE-Net, that approximates the PDE (1) in the way that: 1) we can predict the dynamical behavior of the equation for as long time as possible; 2) we are able to reveal the form of the response function $F$ and the differential operators involved. There are two main components of the PDE-Net that are combined together in the same network: one is automatic determination on the differential operators involved in the PDE and their discrete approximations; the other is to approximate the nonlinear response function $F$. In this section, we start with discussions on the relation between convolutions and differentiations in discrete setting.

### 2.1 CONVOLUTIONS AND DIFFERENTIATIONS

A comprehensive analysis on the relations between convolutions and differentiations within variational and PDE framework were laid out by Cai et al. (2012) and Dong et al. (2017), where the authors established general connections between PDE based approach and wavelet frame based approach for image restoration problems. We demonstrate one of the key observations of their work using a simple example. Consider the 2-dimensional Haar wavelet frame filter bank contains one low-pass filter $h_{00}$ and three high pass filters $h_{10}$, $h_{01}$ and $h_{11}$:

$$h_{00} = \frac{1}{4} \begin{pmatrix} 1 & 1 \\ 1 & 1 \end{pmatrix}, h_{10} = \frac{1}{4} \begin{pmatrix} 1 & -1 \\ 1 & -1 \end{pmatrix}, h_{01} = \frac{1}{4} \begin{pmatrix} 1 & 1 \\ -1 & -1 \end{pmatrix}, h_{11} = \frac{1}{4} \begin{pmatrix} 1 & -1 \\ -1 & 1 \end{pmatrix}.$$

The associated Haar wavelet frame transform on an image $u$ is defined by

$$Wu = \{h_{ij}[-\cdot] \circledast u : 0 \leq i, j \leq 1\},$$

where $\circledast$ is the circular convolution. It is easy to verify using Taylor's expansion that the high frequency coefficients of the Haar wavelet frame transform on $u$ are discrete approximations of differential operators:

$$h_{10}[-\cdot] \circledast u \approx \frac{1}{2}\delta_x u_x, \ h_{01}[-\cdot] \circledast u \approx \frac{1}{2}\delta_y u_y, \ h_{11}[-\cdot] \circledast u \approx \frac{1}{4}\delta_x \delta_y u_{xy}.$$

Here, $\delta_x$ and $\delta_y$ represent the horizontal and vertical spatial grid size respectively. For simplicity of notation, we use regular character to denote both discrete and continuum functions, since there should be no confusion within the context.

A profound relationship between convolutions and differentiations was presented in Dong et al. (2017), where the authors discussed the connection between the order of sum rules of filters and the orders of differential operators. Note that the order of sum rules is closely related to the order of vanishing moments in wavelet theory (Daubechies, 1992; Mallat, 1999). We first recall the definition of the order of sum rules.

**Definition 2.1** (Order of Sum Rules). *For a filter $q$, we say $q$ to have sum rules of order $\alpha = (\alpha_1, \alpha_2)$, where $\alpha \in \mathbb{Z}_+^2$, provided that*

$$\sum_{k \in \mathbb{Z}^2} k^\beta q[k] = 0 \tag{2}$$

*for all $\beta \in \mathbb{Z}_+^2$ with $|\beta| < |\alpha|$ and for all $\beta \in \mathbb{Z}_+^2$ with $|\beta| = |\alpha|$ but $\beta \neq \alpha$. If (2) holds for all $\beta \in \mathbb{Z}_+^2$ with $|\beta| < K$ except for $\beta \neq \beta_0$ with certain $\beta_0 \in \mathbb{Z}_+^2$ and $|\beta_0| = J < K$, then we say $q$ to have total sum rules of order $K \backslash \{J + 1\}$.*

The following proposition from Dong et al. (2017) links the orders of sum rules with orders of differential operator.

**Propositin 2.1.** *Let $q$ be a filter with sum rules of order $\alpha \in \mathbb{Z}_+^2$. Then for a smooth function $F(x)$ on $\mathbb{R}^2$, we have*

$$\frac{1}{\varepsilon^{|\alpha|}} \sum_{k \in \mathbb{Z}^2} q[k] F(x + \varepsilon k) = C_\alpha \frac{\partial^\alpha}{\partial x^\alpha} F(x) + O(\varepsilon), \text{ as } \varepsilon \to 0, \tag{3}$$

*where $C_\alpha$ is the constant defined by*

$$C_\alpha = \frac{1}{\alpha!} \sum_{k \in \mathbb{Z}^2} k^\alpha q[k].$$

*If, in addition, $q$ has total sum rules of order $K \backslash \{|\alpha| + 1\}$ for some $K > |\alpha|$, then*

$$\frac{1}{\varepsilon^{|\alpha|}} \sum_{k \in \mathbb{Z}^2} q[k] F(x + \varepsilon k) = C_\alpha \frac{\partial^\alpha}{\partial x^\alpha} F(x) + O(\varepsilon^{K - |\alpha|}), \text{ as } \varepsilon \to 0. \tag{4}$$

According to Proposition 2.1, an $\alpha$th order differential operator can be approximated by the convolution of a filter with $\alpha$ order of sum rules. Furthermore, according to (4), one can obtain a high order approximation of a given differential operator if the corresponding filter has an order of total sum rules with $K > |\alpha| + k, k \geqslant 1$. For example, the filter $h_{10}$ in the Haar wavelet frame filter bank has a sum rules of order $(1, 0)$, and a total sum rules of order $2 \backslash \{2\}$. Thus, up to a constant and a proper scaling, $h_{10}$ corresponds to a discretization of $\frac{\partial}{\partial x}$ with first order approximation. The filer $h_{11}$ has a sum rules of order $(1, 1)$, and a total sum rules of order $3 \backslash \{3\}$. Thus, up to a constant and a proper scaling, $h_{11}$ corresponds to a discretization of $\frac{\partial^2}{\partial x \partial y}$ with first order approximation. Finally, consider filter

$$q = \begin{pmatrix} 1 & 0 & -1 \\ 2 & 0 & -2 \\ 1 & 0 & -1 \end{pmatrix}.$$

It has a sum rules of order $(1, 0)$, and a total sum rules of order $3 \backslash \{2\}$. Thus, up to a constant and a proper scaling, $q$ corresponds to a discretization of $\frac{\partial}{\partial x}$ with second order approximation.

Now, we introduce the concept of *moment matrix* for a given filter that will be used to constrain filters in the PDE-Net. For an $N \times N$ filter $q$, define the moment matrix of $q$ as

$$M(q) = (m_{i,j})_{N \times N}, \text{ where } m_{i,j} = \frac{1}{(i-1)!(j-1)!} \sum_{k \in \mathbb{Z}^2} k_1^{i-1} k_2^{j-1} q[k_1, k_2], \tag{5}$$

for $i, j = 1, 2, \ldots, N$. We shall call the $(i, j)$-element of $M(q)$ the $(i - 1, j - 1)$-moment of $q$ for simplicity. Combining (5) and Proposition 2.1, one can easily see that filter $q$ can be designed to approximate any differential operator at any given approximation order by imposing constraints on $M(q)$. For example, if we want to approximate $\frac{\partial u}{\partial x}$ (up to a constant) by convolution $q \circledast u$ where $q$ is a $3 \times 3$ filter, we can consider the following constrains on $M(q)$:

$$\begin{pmatrix} 0 & 0 & \star \\ 1 & \star & \star \\ \star & \star & \star \end{pmatrix} \quad \text{or} \quad \begin{pmatrix} 0 & 0 & 0 \\ 1 & 0 & \star \\ 0 & \star & \star \end{pmatrix}. \tag{6}$$

Here, $\star$ means no constraint on the corresponding entry. The constraints described by the moment matrix on the left of (6) guarantee the approximation accuracy is at least first order, and the ones on

the right guarantee an approximation of at least second order. In particular, when all entries of $M(q)$ are constrained, e.g.

$$M(q) = \left( \begin{array}{ccc} 0 & 0 & 0 \\ 1 & 0 & 0 \\ 0 & 0 & 0 \end{array} \right),$$

the corresponding filter can be uniquely determined, in which case we call it a "frozen" filter. In the PDE-Net which shall be introduced in the next subsection, all filters are learned subjected to partial constraints on their associated moment matrices.

It is worth noticing that the approximation property of a filter is limited by its size. Generally speaking, large filters can approximate higher order differential operators or lower order differential operators with higher approximation orders. Taking 1-dimensional case as an example, 3-element filters cannot approximate the fifth order differential operator, whereas 7-element filters can. In other words, the larger are the filters, the stronger is the representation capability of filters. However, larger filters lead to more memory overhead and higher computation cost. It is a wisdom to balance the trade-off in practice.

## 2.2 Architecture of PDE-Net

Given the evolution PDE (1), we consider forward Euler as the temporal discretization. One may consider more sophisticated temporal discretization which leads to different network architectures. For simplicity, we focus on forward Euler in this paper.

**One $\delta t$-Block:**

Let $\tilde{u}(t_{i+1}, \cdot)$ be the predicted value of $u$ at time $t_{i+1}$ based on the value of $u$ at $t_i$. Then, we have

$$\tilde{u}(t_{i+1}, \cdot) = D_0 u(t_i, \cdot) + \Delta t \cdot F(x, y, D_{00}u, D_{10}u, D_{01}u, D_{20}u, D_{11}u, D_{02}u, \ldots). \tag{7}$$

Here, the operators $D_0$ and $D_{ij}$ are convolution operators with the underlying filters denoted by $q_0$ and $q_{ij}$, i.e. $D_0 u = q_0 \circledast u$ and $D_{ij} u = q_{ij} \circledast u$. The operators $D_{10}$, $D_{01}$, $D_{11}$, etc. approximate differential operators, i.e. $D_{ij} u \approx \frac{\partial^{i+j} u}{\partial^i x \partial^j y}$. The operators $D_0$ and $D_{00}$ are average operators. The purpose of introducing these average operators in stead of using the identity is to improve stability of the network and enables it to capture more complex dynamics. Other than the assumption that the observed dynamics is governed by a PDE of the form (1), we assume that the highest order of the PDE is less than some positive integer. Then, the task of approximating $F$ is equivalent to a multivariate regression problem, which can be approximated by a point-wise neural network (with shared weights across the computation domain $\Omega$) or other classical machine learning methods. Combining the approximation of differential operators and the nonlinear function $F$, we achieve an approximation framework of (7) which will be referred to as a $\delta t$-block (see Figure 1). Note that if we have a prior knowledge on the form of the response function $F$, we can easily adjust the network architecture by taking advantage of the additional information. This may simplify the training and improve the results.

**PDE-Net (Multiple $\delta t$-Blocks):**

One $\delta t$-block only guarantees the accuracy of one-step dynamics, which does not take error accumulation into consideration. This may cause severe instability in prediction. To improve the stability of the network and enable long-term prediction, we stack multiple $\delta t$-blocks into a deep network, and call this network the *PDE-Net* (see Figure 2). The importance of stacking multiple $\delta t$-blocks will be demonstrated in Section 3.

The PDE-Net can be easily described as: (1) stacking one $\delta t$-block multiple times; (2) sharing parameters in all $\delta t$-blocks. Given an input data $u(t_i, \cdot)$, training a PDE-Net with $n$ $\delta t$-blocks needs to minimize the accumulated error $\|u(t_{i+n}, \cdot) - \tilde{u}(t_{i+n}, \cdot)\|_2^2$, where $\tilde{u}(t_{i+n}, \cdot)$ is the output from the PDE-Net (i.e. $n$ $\delta t$-blocks) with input $u(t_i, \cdot)$. Thus, the PDE-Net with bigger $n$ owns a longer time stability. Note that sharing parameters is a common practice in deep learning, which decreases the number of parameters and leads to significant memory reduction (Goodfellow et al., 2016).

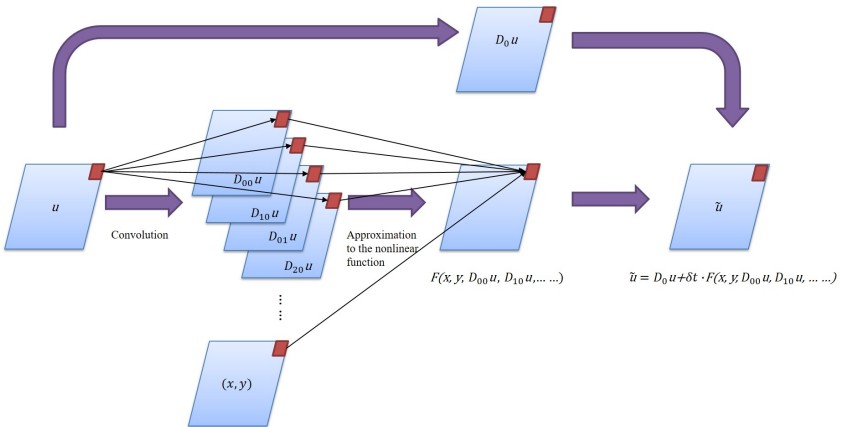

Figure 1: The schematic diagram of a $\delta t$-block.

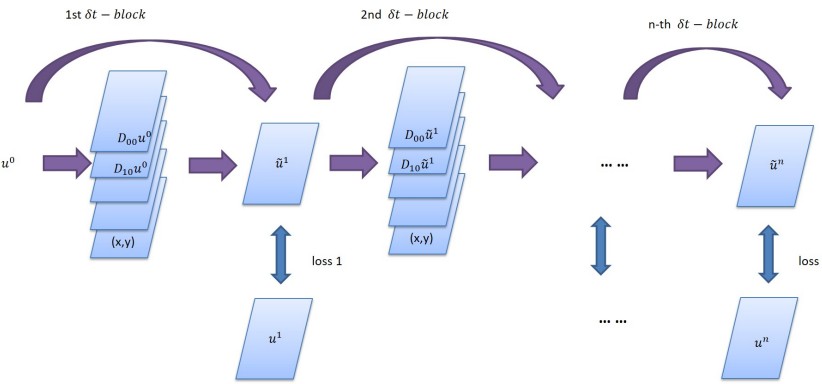

Figure 2: The schematic diagram of the PDE-Net: multiple $\delta t$-blocks.

### LOSS FUNCTION AND CONSTRAINTS:

Consider the data set $\{u_j(t_i, \cdot) : i, j = 0, 1, \ldots\}$, where $j$ indicates the $j$-th solution path with a certain initial condition of the unknown dynamics. We would like to train the PDE-Net with $n$ $\delta t$-blocks. For a given $n \geq 1$, every pair of the data $\{u_j(t_i, \cdot), u_j(t_{i+n}, \cdot)\}$, for each $i$ and $j$, is a training sample, where $u_j(t_i, \cdot)$ is the input and $u_j(t_{i+n}, \cdot)$ is the label that we need to match with the output from the PDE-Net. We select the following simple $\ell_2$ loss function for training:

$$L = \sum_{i,j} l_{ij}, \text{where} \quad l_{ij} = ||u_j(t_{i+n}, \cdot) - \tilde{u}_j(t_{i+n}, \cdot)||_2^2,$$

where $\tilde{u}_j(t_{i+n}, \cdot)$ is the output of the PDE-Net with $u_j(t_i, \cdot)$ as the input.

All the filters involved in the PDE-Net are properly constrained using their associated moment matrices. Let $q_0$ and $q_{ij}$ be the underlying filters of $D_0$ and $D_{ij}$. We impose the following constrains

$$(M(q_0))_{1,1} = 1, \quad (M(q_{00}))_{1,1} = 1$$

and for $i + j > 0$

$$\begin{cases} (M(q_{i,j}))_{k_1,k_2} = 0 & k_1 + k_2 \leq i + j + 2, \ (k_1, k_2) \neq (i+1, j+1), \\ (M(q_{i,j}))_{k_1,k_2} = 1 & (k_1, k_2) = (i+1, j+1). \end{cases}$$

For example, for $3 \times 3$ filters, we have

$$M(q_0) = M(q_{00}) = \begin{pmatrix} 1 & \star & \star \\ \star & \star & \star \\ \star & \star & \star \end{pmatrix}$$

and

$$M(q_{10}) = \begin{pmatrix} 0 & 0 & \star \\ 1 & \star & \star \\ \star & \star & \star \end{pmatrix}, \ M(q_{01}) = \begin{pmatrix} 0 & 1 & \star \\ 0 & \star & \star \\ \star & \star & \star \end{pmatrix}, \ M(q_{11}) = \begin{pmatrix} 0 & 0 & 0 \\ 0 & 1 & \star \\ 0 & \star & \star \end{pmatrix}, \ldots .$$

To demonstrate the necessity of learnable filters, we will compare the PDE-Net having the afore-mentioned constrains on the filters with the PDE-Net having frozen filters. To differentiate the two cases, we shall call the PDE-Net with frozen filters "the Frozen-PDE-Net".

To further increase the expressive power and flexibility of the PDE-Net, we may associate multiple filters to approximate a given differential operator. However, in order not to mess up the identifiability of the underlying PDE model, we may select only one of the filters to provide correct approximation to the given differential operator in the way as described above. The rest of the filters are constrained in the way that they only contribute to modify the local truncation errors. For example, consider two $3 \times 3$ filters $\{q_0, q_1\}$ and constrain their moment matrices as follows

$$M(q_0) = \begin{pmatrix} 0 & 0 & \star \\ 1 & \star & \star \\ \star & \star & \star \end{pmatrix}, \ M(q_1) = \begin{pmatrix} 0 & 0 & \star \\ 0 & \star & \star \\ \star & \star & \star \end{pmatrix}.$$

Then, $q_0 \circledast u + q_1 \circledast u$ is potentially a better approximation to $u_x$ (up to a constant) than $q_0 \circledast u$. However, for simplicity, we only use one filter to approximate a given differential operator in this paper.

**NOVELTY OF THE PDE-NET:**

Different from fixing numerical approximations of differentiations in advance in sparse regression methods (Schaeffer, 2017; Rudy et al., 2017), using learnable filters makes the PDE-Net more flexible, and enables more robust approximation of unknown dynamics and longer time prediction (see numerical experiments in Section 3 and Section 4). Furthermore, the specific form of the response function $F$ is also approximated from the data, rather than assumed to be known in advance (such as (Raissi & Karniadakis, 2017)). On the other hand, by inflicting constrains on moment matrices, we can identify which differential operators are included in the underlying PDE which helps with identifying the nonlinear response function $F$. This grants transparency to the PDE-Net and the potential to reveal hidden physical laws. Therefore, the proposed PDE-Net is distinct from the existing learning based method to discover PDEs from data, as well as networks designed in deep learning for computer vision tasks.

### 2.3 INITIALIZATION AND TRAINING

In the PDE-Net, parameters can be divided into three groups:

- filters to approximate differential operators;
- the parameters of the point-wise neural network to approximate $F$;
- hyper-parameters, such as the number of filters, the size of filters, the number of layers, etc.

The parameters of the point-wise neural network are shared across the computation domain $\Omega$, and are initialized by random sampling from a Gaussian distribution. For the filters, we initialize them by freezing them to their corresponding differential operators. For example, if a filter is to approximate $\frac{\partial}{\partial x}$, we freeze it by constraining its $(1, 0)$-moment to 1 and other moments to 0. During the training process, we release the filters by switching to the constrains described in Section 2.2.

Instead of training an $n$-layer PDE-Net directly, we adopt layer-wise training, which improves the training speed. To be more precise, we start with training the PDE-Net on the first $\delta t$-block, and then use the results of the first $\delta t$-block as the initialization and restart training the PDE-Net on the first two $\delta t$-blocks. Repeat until we complete all $n$ blocks. Note that all the parameters in each of the $\delta t$-block are shared across layers. In addition, we add a warm-up step before the training of the first $\delta t$-block. The warm-up step is to obtain a good initial guess of the parameters of the point-wise neural network that approximates $F$ by using frozen filters.

## 2.4 RELATIONS TO SOME EXISTING NETWORKS

In recent years, a variety of deep neural networks have been introduced with great success in computer vision. The structure of the proposed PDE-Net is similar to some existing networks such as the Network-In-Network (NIN) (Lin et al., 2013) and the deep Residual Neural Network (ResNet) (He et al., 2016a;b).

The NIN is an improvement over the traditional convolutional neural networks. One of the special designs of NIN is the use of multilayer perceptron convolution (mlpconv) layers instead of the ordinary convolution layers. An mlpconv layer contains the convolutions and small point-wise neural networks. Such design can improve the ability of the network to extract nonlinear features from shallow layers. The inner structure of one $\delta t$-block of the PDE-Net is similar to the mlpconv layer, and the multiple $\delta t$-blocks structure is similar to the NIN structure, except for the pooling and ReLU operations.

On the other hand, each $\delta t$-block of the PDE-Net has two paths (see Figure 1 and Figure 2): one is for the averaged quantity of $u$ and the other is for the increment $F$. This structure coincides with the "residual block" introduced in ResNet. In fact, there has been a substantial study on the relation between ResNet and dynamical systems recently (E, 2017; Haber & Ruthotto, 2017; Sonoda & Murata, 2017).

# 3 NUMERICAL STUDIES: CONVECTION-DIFFUSION EQUATIONS

Convection-diffusion equations are classical PDEs that are used to describe physical phenomena where particles, energy, or other physical quantities are transferred inside a physical system due to two processes: diffusion and convection (Chandrasekhar, 1943). Convection-diffusion equations are widely applied in many scientific areas and industrial fields, such as pollutants dispersion in rivers or atmosphere, solute transferring in a porous medium, and oil reservoir simulation. In practical situations, usually the physical and chemical properties on different locations cannot be the same (called anisotropy in physics), thus it is more reasonable that convection coefficients and diffusion coefficients are variables instead of constants.

## 3.1 SIMULATED DATA, TRAINING AND TESTING

We consider a 2-dimensional linear variable-coefficient convection-diffusion equation on $\Omega = [0, 2\pi] \times [0, 2\pi]$,

$$\begin{cases} \frac{\partial u}{\partial t} &= a(x,y)u_x + b(x,y)u_y + cu_{xx} + du_{yy} \\ u|_{t=0} &= u_0(x,y), \end{cases} \quad \text{with } (t,x,y) \in [0,0.2] \times \Omega, \quad (8)$$

where

$$a(x,y) = 0.5(\cos(y) + x(2\pi - x)\sin(x)) + 0.6, \quad b(x,y) = 2(\cos(y) + \sin(x)) + 0.8,$$

$c = 0.2$ and $d = 0.3$.

The computation domain $\Omega$ is discretized using a $50 \times 50$ regular mesh. Data is generated by solving problem (8) using a high precision numerical scheme with pseudo-spectral method for spatial discretization and 4th order Runge-Kutta for temporal discretization (with time step size $\delta t = 0.01$). We assume periodic boundary condition and the initial value $u_0(x,y)$ is generated from

$$u_0(x,y) = \sum_{|k|,|l| \leq N} \lambda_{k,l} \cos(kx + ly) + \gamma_{k,l} \sin(kx + ly), \quad (9)$$

where $N = 9$, $\lambda_{k,l}, \gamma_{k,l} \sim \mathcal{N}(0, \frac{1}{50})$, and $k$ and $l$ are chosen randomly. In order to mimic real world scenarios, we add noise to the generated data. For each sample sequence $u(x,y,t)$, $t \in [0, 0.2]$, the noise is added as

$$\widehat{u}(x,y,t) = u(x,y,t) + 0.01 \times MW \quad (10)$$

where $M = \max_{x,y,t}\{u(x,y,t)\}$, $W \sim \mathcal{N}(0,1)$ and $\mathcal{N}(0,1)$ represents the standard normal distribution.

Suppose we know a priori that the underlying PDE is linear with order no more than 4. Then, the response function $F$ takes the following form

$$F = \sum_{0 \leq i+j \leq 4} f_{ij}(x,y) \frac{\partial^{i+j} u}{\partial x^i \partial y^j}.$$

Each $\delta t$-block of the PDE-Net can be written as

$$\tilde{u}(t_{n+1}, \cdot) = D_0 u(t_n, \cdot) + \delta t \cdot (c_{00} D_{00} u + c_{10} D_{10} u + \ldots + c_{04} D_{04} u),$$

where $\{D_0, D_{ij} : i + j \leq 4\}$ are convolution operators and $\{c_{ij} : i + j \leq 4\}$ are 2D arrays which approximate functions $f_{ij}(x,y)$ on $\Omega$. The approximation is achieved using piecewise quadratic polynomial interpolation with smooth transitions at the boundaries of each piece. The filters associated to the convolution operators $\{D_0, D_{ij} : i + j \leq 4\}$ and the coefficients of the piecewise quadratic polynomials are the trainable parameters of the network.

During training and testing, the data is generated on-the-fly, i.e. we only generate the data needed following the aforementioned procedure when training and testing the PDE-Net. In our experiments, the size of the filters that will be used is $5 \times 5$ or $7 \times 7$. The total number of trainable parameters for each $\delta t$-block is approximately 17k. During training, we use LBFGS, instead of SGD, to optimize the parameters. We use 28 data samples per batch to train each layer (i.e. $\delta t$-block) and we only construct the PDE-Net up to 20 layers, which requires totally 560 data samples per batch. Note that the PDE-Net is designed with the assumption that it approximates nonlinear evolution PDEs, which is a relatively stronger assumption than the networks in deep learning. Therefore, we require less training data and LBFGS performs better than SGD (which is widely adopted in deep learning). Furthermore, as will be shown by our numerical results, the learned PDE-Net generalizes very well. The PDE-Net can accurately predict the dynamics even when the initial data $u_0$ does not come from the same distribution as in the training process.

## 3.2 RESULTS AND DISCUSSIONS

This section presents numerical results of training the PDE-Net using the data set described in the previous subsection. We will specifically observe how the learned PDE-Net performs in terms of prediction of dynamical behavior and identification of the underlying PDE model. Furthermore, we will investigate the effects of some of the hyper-parameters (e.g. size of the filters, number of $\delta t$-blocks) on the learned PDE-Net.

### PREDICTING LONG-TIME DYNAMICS

We demonstrate the ability of the trained PDE-Net in prediction, which in the language of machine learning is the ability to generalize. After the PDE-Net with $n$ $\delta t$-blocks ($1 \leq n \leq 20$) is trained, we randomly generate 560 initial guesses based on (9) and (10), feed them to the PDE-Net, and measure the normalized error between the predicted dynamics (i.e. the output of the PDE-Net) and the actual dynamics (obtained by solving (8) using high precision numerical scheme). The normalized error between the true data $u$ and the predicted data $\tilde{u}$ is defined as

$$\epsilon = \frac{\|\tilde{u} - u\|_2^2}{\|u - \bar{u}\|_2^2},$$

where $\bar{u}$ is the spatial average of $u$. The error plots are shown in Figure 3. Results of longer prediction for the PDE-Net with $7 \times 7$ learnable filters are shown in Figure 4. Some of the images of the predicted dynamics are presented in Figure 5. From these results, we can see that:

- Even trained with noisy data, the PDE-Net is able to perform long-term prediction (see Figure 5);

- Having multiple $\delta t$-blocks helps with the stability of the PDE-Net and ensures long-term prediction (see Figure 3);

- The PDE-Net performs significantly better than Frozen-PDE-Net, especially for $7 \times 7$ filters (see Figure 3);

- The PDE-Net with $7 \times 7$ filters significantly outperforms the PDE-Net with $5 \times 5$ filters in terms of the length of reliable predictions (see Figure 3 and 4). To reach an $O(1)$ error, the length of prediction for the PDE-Net with $7 \times 7$ filters is about 10 times of that for the PDE-Net with $5 \times 5$ filters.

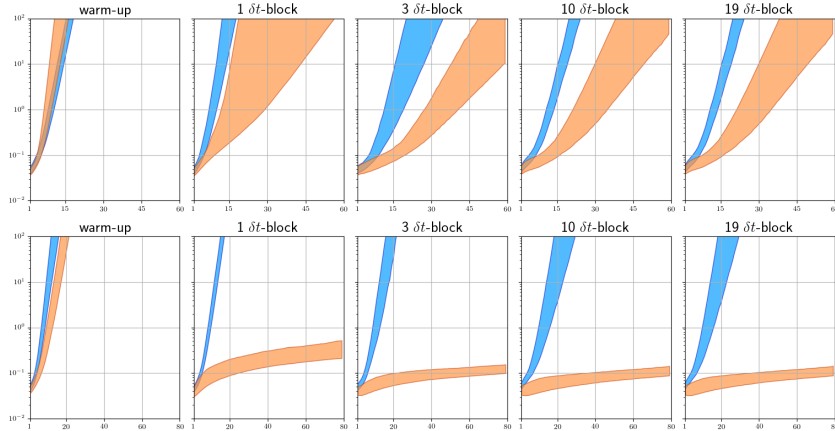

Figure 3: Prediction errors of the PDE-Net (orange) and Frozen-PDE-Net (blue) with $5 \times 5$ (first row) and $7 \times 7$ (second row) filters. In each plot, the horizontal axis indicates the time of prediction in the interval $(0, 60 \times \delta t] = (0, 0.6]$, and the vertical axis shows the normalized errors. The banded curves indicate the 25% & 75% percentile of the normalized errors among 560 test samples.

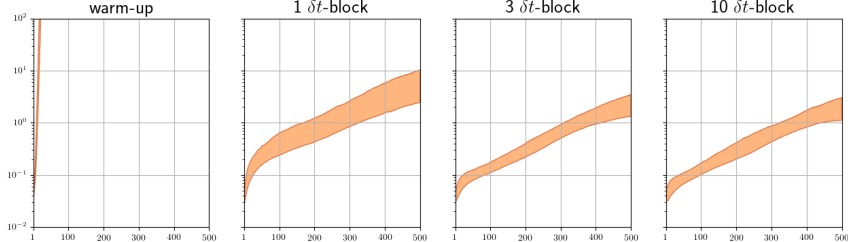

Figure 4: Long-time prediction for the PDE-Net with $7 \times 7$ filters. The horizontal axis ranges in $(0, 5]$. Time step $\delta t = 0.01$.

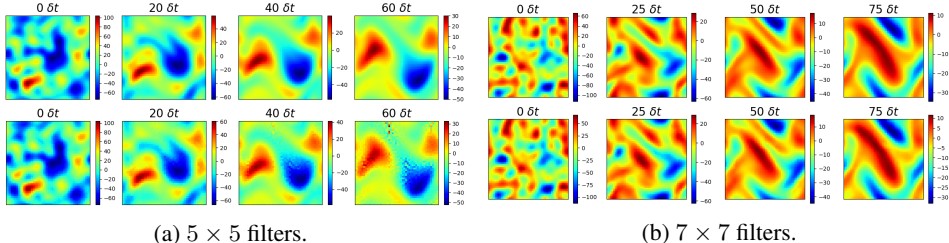

(a) $5 \times 5$ filters.               (b) $7 \times 7$ filters.

Figure 5: Images of the true dynamics and the predicted dynamics. The first row shows the images of the true dynamics. The second row shows the images of the predicted dynamics using the PDE-Net having 3 $\delta t$-blocks with $5 \times 5$ and $7 \times 7$ filters. Time step $\delta t = 0.01$.

DISCOVERING THE HIDDEN EQUATION

For the linear problem, identifying the PDE amounts to finding the coefficients $\{c_{ij} : i + j \leq 4\}$ that approximate $\{f_{ij} : i + j \leq 4\}$. The coefficients $\{c_{ij} : i + j \leq 2\}$ of the trained PDE-Net are

shown in Figure 6. Note that $\{f_{11}\} \cup \{f_{ij} : 2 < i + j \le 4\}$ are absent from the PDE (8), and the corresponding coefficients learned by the PDE-Net are indeed close to zero. In order to have a more concise demonstration of the results, we only show the image of $\{c_{ij} : i + j \le 2\}$ in Figure 6.

Comparing the first three rows of Figure 6, the coefficients $\{c_{ij}\}$ learned by the PDE-Net are close to the true coefficients $\{f_{ij}\}$ except for some oscillations due to the presence of noise in the training data. Furthermore, the last row of Figure 6 indicates that having multiple $\delta t$-blocks helps with estimation of the coefficients. However, having larger filters does not seem to improve the learning of the coefficients, though it helps tremendously in prolonging predictions of the PDE-Net.

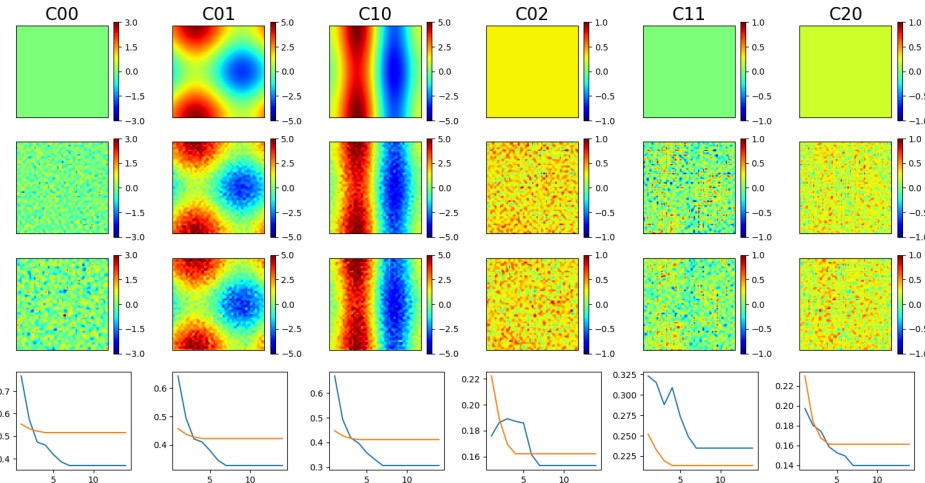

Figure 6: First row: the true coefficients of the equation. From the left to right are coefficients of $u$, $u_x$, $u_y$, $u_{xx}$, $u_{xy}$ and $u_{yy}$. Second row: the learned coefficients by the PDE-Net with 6 $\delta t$-blocks and $5 \times 5$ filters. Third row: the learned coefficients by the PDE-Net with 6 $\delta t$-blocks and $7 \times 7$ filters. Last row: the errors between true and learned coefficients v.s. number of $\delta t$-blocks $(1, 2, \ldots, 13)$ with different sizes of filters (blue for $5 \times 5$ and orange for $7 \times 7$).

FURTHER EXPERIMENTS

To further demonstrate how well the learned PDE-Net generalizes, we generate initial values following (9) with highest frequency equal to 12, followed by adding noise (10). Note that the maximum allowable frequency in the training set is 9. The results of long-time prediction and the estimated dynamics are shown in Figure 7. Although oscillations are observed in the prediction, the estimated dynamic still captures the main pattern of the true dynamic.

The PDE (8) is of second order. In our previous experiments, we assumed that the PDE does not exceed the 4th order. If we know that the PDE is of second order, we will be able to have a more accurate estimation of the variable coefficients of the convection and diffusion terms. However, the prediction errors are slightly higher since we have fewer trainable parameters. Nonetheless, since we are using a more accurate prior knowledge on the unknown PDE, the variance of the prediction errors are smaller than before. These results are summarized in Figure 8 (green curves) and Figure 9.

To further demonstrate the importance of the moment constraints on the filters in the PDE-Net, we trained the network without any moment constraints and skipped any steps that utilize the knowledge of the relation between the filters and differential operators (i.e. we skipped warm-up and the initialization using finite difference filters). For simplicity, we call the PDE-Net train in this way as the Freed-PDE-Net. The prediction errors of the Freed-PDE-Net are shown as the red curves in Figure 8. Since without moment constraints, we do not know the correspondence of the filters with differential operators. Therefore, we cannot identify the correspondence of the learned variable coefficients either. We plot all the 15 variable coefficients (assuming the underlying PDE is of order $\le 4$) in Figure 10. As one can see that the Freed-PDE-Net is better in prediction than the PDE-Net

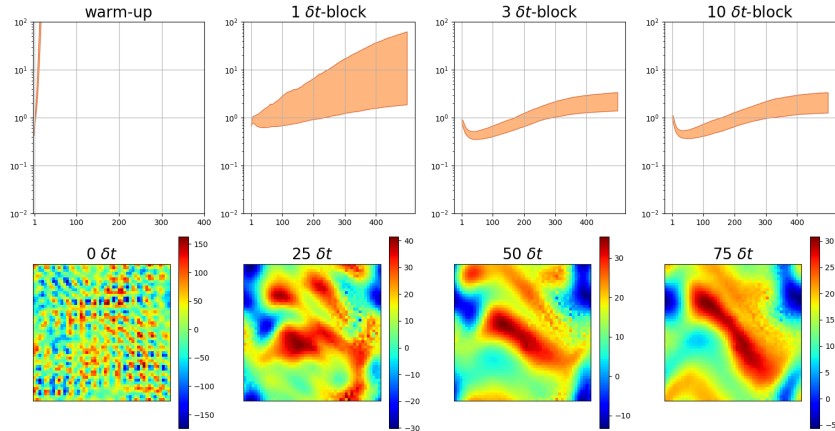

Figure 7: Testing with higher frequency initializations (linear convection-diffusion equation). First row: long-time prediction. Second row: estimated dynamics. Here, $\delta t = 0.01$.

since it has more trainable parameters than the PDE-Net. However, we are unable to identify the PDE from the Free-PDE-Net.

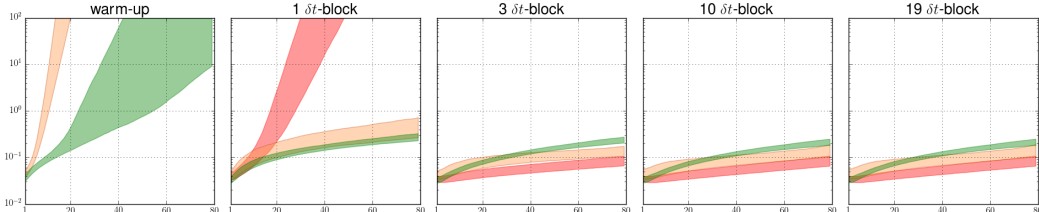

Figure 8: Prediction errors of the PDE-Net assuming the underlying PDE has order $\leq 4$ (orange), order $\leq 2$ (green) and Freed-PDE-Net (red) with $7 \times 7$ filters. In each plot, the horizontal axis indicates the time of prediction in the interval $(0, 80 \times \delta t] = (0, 0.8]$, and the vertical axis shows the normalized errors. The banded curves indicate the 25% & 75% percentile of the normalized errors among 560 test samples.

QUICK SUMMARY:

In summary, the numerical experiments show that the PDE-Net is able to conduct accurate prediction and identify the underlying PDE model at the same time, even in a noisy environment. Multiple $\delta t$-blocks, i.e. deeper structure of the PDE-Net, makes the PDE-Net more stable and enables longer time prediction. Furthermore, using larger filters helps with stability and can prolong reliable predictions. Comparisons of the PDE-Net with the Frozen-PDE-Net and Freed-PDE-Net demonstrate the importance of using learnable and yet partially constrained filters, which is new to the literature.

## 4 NUMERICAL STUDIES: DIFFUSION EQUATIONS WITH NONLINEAR SOURCE

When modeling physical processes like particle transportation or energy transfer, in addition to convection and diffusion, we have to consider source/sink terms. In some problems, the source/sink plays an important role. For example, when convection-diffusion equations are used to describe the distribution and flow of pollutants in water or atmosphere, identifying the intensity of pollution source is equivalent to finding the source term, which is important for environmental pollution control problems.

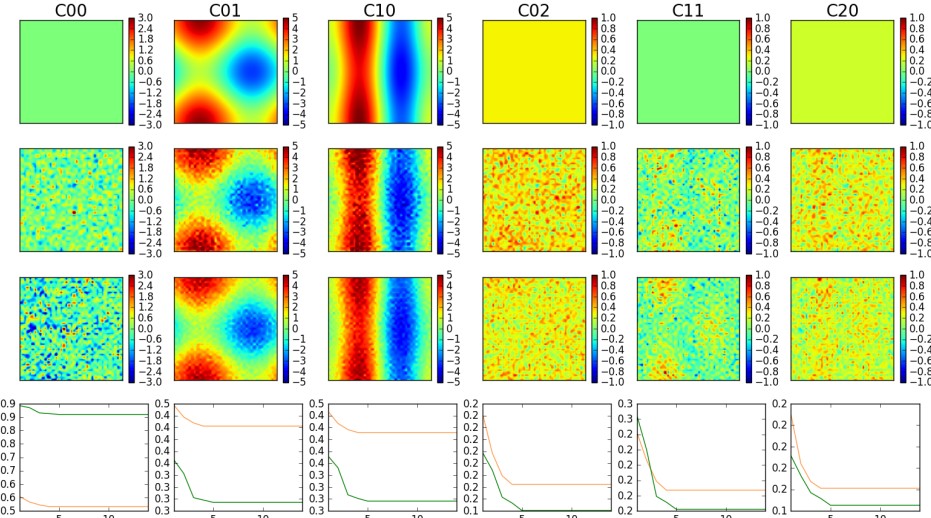

Figure 9: First row: the true coefficients of the equation. From the left to right are coefficients of $u$, $u_x$, $u_y$, $u_{xx}$, $u_{xy}$ and $u_{yy}$. Second row: the learned coefficients by the PDE-Net assuming the order of the PDE is $\leq 4$ (same as the third row of Figure 6). Third row: the learned coefficients by the PDE-Net assuming the order of the PDE is $\leq 2$. Last row: the errors between true and learned coefficients v.s. number of $\delta t$-blocks $(1, 2, \ldots, 13)$ for PDE-Net assuming the PDE is of order $\leq 4$ (orange) and $\leq 2$ (green).

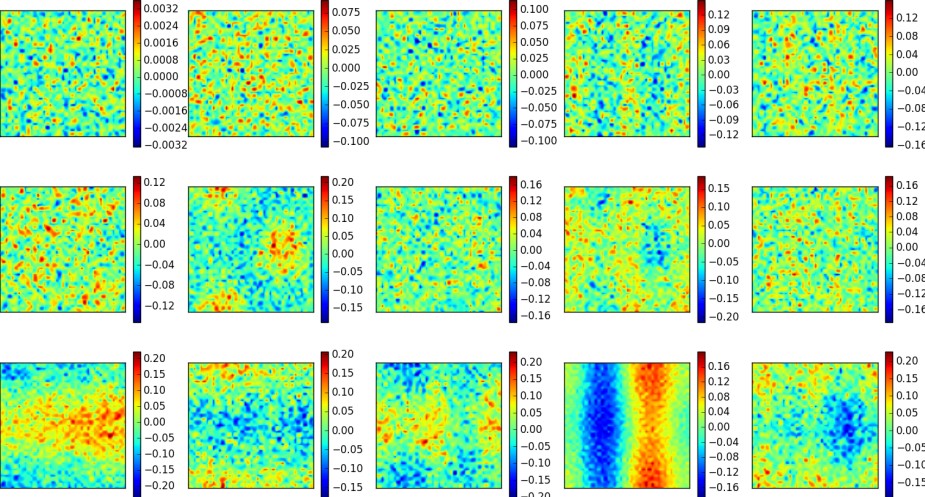

Figure 10: The images of all the variable coefficients learned from the Freed-PDE-Net.

## 4.1 SIMULATED DATA, TRAINING AND TESTING

We consider a 2-dimensional linear diffusion equation with a nonlinear source on $\Omega = [0, 2\pi] \times [0, 2\pi]$,

$$\begin{cases} \frac{\partial u}{\partial t} & = c\Delta u + f_s(u) \\ u|_{t=0} & = u_0(x, y), \end{cases} \quad \text{with } (t, x, y) \in [0, 0.2] \times \Omega, \tag{11}$$

where $c = 0.3$ and $f_s(u) = 15\sin(u)$. The computation domain $\Omega$ is discretized using a $50 \times 50$ regular mesh. Data is generated by solving problem (11) using forward Euler for temporal discretization (with time step size $\delta t = 0.0009$) and central differencing for spatial discretization on $100 \times 100$ mesh, and then restricted to the $50 \times 50$ mesh. We assume zero boundary condition and

the initial value $u_0(x, y)$ is generated by $u_0(x, y) = u'_0(x, y)\frac{x(2\pi-x)y(2\pi-y)}{(2\pi)^4}$, where $u'_0$ is obtained from (9) with maximum allowable frequency $N = 6$. Same as the numerical setting in Section 3, Gaussian noise is added to each sample sequence $u(x, y, t)$, $t \in [0, 0.2]$ as described by (10).

Suppose we know a priori that the underlying PDE is a convection-diffusion equation of order no more than 2 with a nonlinear source depending on the variable $u$. Then, the response function $F$ takes the following form

$$F = \sum_{1 \leq i+j \leq 2} f_{ij}(x, y)\frac{\partial^{i+j}u}{\partial x^i \partial y^j} + f_s(u).$$

Each $\delta t$-block of the PDE-Net can be written as

$$\tilde{u}(t_{n+1}, \cdot) = D_0 u(t_n, \cdot) + \delta t \cdot (c_{01}D_{01}u + c_{10}D_{10}u + c_{11}D_{11}u + c_{20}D_{20}u + c_{02}D_{02}u) + \tilde{f}_s(u),$$

where $\{D_0, D_{ij} : 1 \leq i + j \leq 2\}$ are convolution operators and $\{c_{ij} : 1 \leq i + j \leq 2\}$ are 2D arrays which approximate functions $f_{ij}(x, y)$ on $\Omega$. The approximation is achieved using piecewise quadratic polynomial interpolation with smooth transitions at the boundaries of each piece. The approximation of $\tilde{f}_s$ is obtained by piecewise 4th order polynomial approximation over a regular grid of the interval $[-30, 30]$ with 40 grid points. The training and testing strategy is exactly the same as in Section 3. In our experiments, the size of the filters is $7 \times 7$. The total number of trainable parameters for each $\delta t$-block is approximately 1.2k.

## 4.2 Results and Discussions

This section presents numerical results of the trained PDE-Net using the data set described in Section 4.1. We will observe how the trained PDE-Net performs in terms of prediction of dynamical behavior and identification of the underlying PDE model.

### Predicting long-time dynamics

We demonstrate the ability of the trained PDE-Net in prediction, which in the language of machine learning is the ability to generalize. The testing method is exactly the same as the method described in Section 3. Comparisons between PDE-Net and Frozen-PDE-Net are shown in Figure 11, where we can clearly see the advantage of learning the filters. Long-time predictions of the PDE-Net is shown in Figure 12 and we visualize the predicted dynamics in Figure 13. To further demonstrate how well the learned PDE-Net generalizes, we generate initial values following (9) with highest frequency equal to 10, followed by adding noise (10). Note that the maximum allowable frequency in the training set is only 6. The results of long-time prediction and the estimated dynamics are shown in Figure 14. All these results show that the learned PDE-Net performs well in prediction.

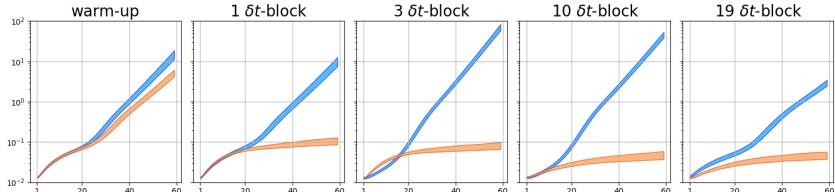

Figure 11: Prediction errors of the PDE-Net (orange) and Frozen-PDE-Net (blue) with $7 \times 7$ filters. In each plot, the horizontal axis indicates the time of prediction in the interval $(0, 0.6]$, and the vertical axis shows the normalized errors. The banded curves indicate the 25% & 75% percentile of the normalized errors among 560 test samples.

### Discovering the hidden equation

For the PDE (11), identifying the PDE amounts to finding the coefficients $\{c_{ij} : 1 \leq i + j \leq 2\}$ that approximate $\{f_{ij} : 1 \leq i + j \leq 2\}$, and $\tilde{f}_s$ that approximates $f_s$. The computed coefficients $\{c_{ij} : 1 \leq i + j \leq 2\}$ of the trained PDE-Net are shown in Figure 15, and the computed $\tilde{f}_s$ is

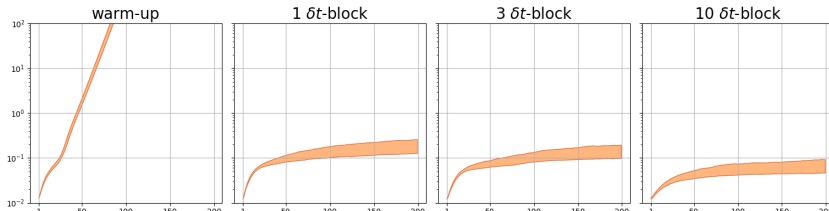

Figure 12: Long-time prediction for the PDE-Net with $7 \times 7$ filters in $(0, 2]$.

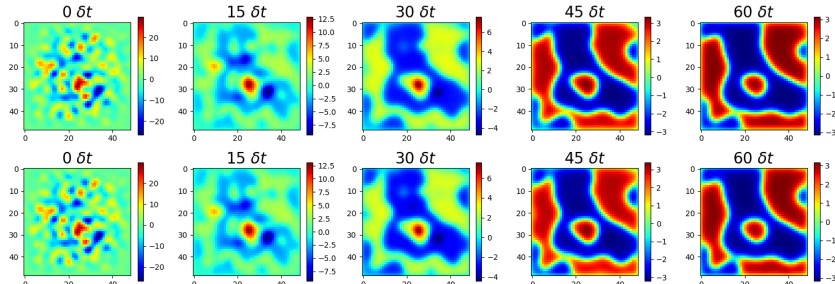

Figure 13: Images of the true dynamics and the predicted dynamics. The first row shows the images of the true dynamics. The second row shows the images of the predicted dynamics using the PDE-Net having 3 $\delta t$-blocks with $7 \times 7$ filters. Here, $\delta t = 0.01$.

shown in Figure 16 (left). Note that the first order terms are absent from the PDE (11), and the corresponding coefficients learned by the PDE-Net are indeed close to zero. The approximation of $f_s$ is more accurate near the center of the interval than near the boundary. This is because the value of $u$ in the data set is mostly distributed near the center (Figure 16(right)).

## 5   CONCLUSION AND DISCUSSION

In this paper, we designed a deep feed-forward network, called the PDE-Net, to discover the hidden PDE model from the observed dynamics and to predict the dynamical behavior. The PDE-Net consists of two major components which are jointly trained: to approximate differential operations by convolutions with properly constrained filters, and to approximate the nonlinear response by deep neural networks or other machine learning methods. The PDE-Net is suitable for learning PDEs as general as in (1). However, if we have a prior knowledge on the form of the response function $F$, we can easily adjust the network architecture by taking advantage of the additional information. This may simplify the training and improve the results. As an example, we considered a linear variable-coefficient convection-diffusion equation. The results show that the PDE-Net can uncover the hidden equation of the observed dynamics, and predict the dynamical behavior for a relatively long time, even in a noisy environment. Furthermore, having deep structure (i.e. multiple $\delta t$-blocks) and larger learnable filters can improve the PDE-Net in terms of stability and can prolong reliable predictions. As part of the future work, we will try the proposed framework on real data sets. One of the important directions is to uncover hidden variables which cannot be measured by sensors directly, such as in data assimilation. Another interesting direction which is worth exploring is to learn stable and consistent numerical schemes for a given PDE model based on the architecture of the PDE-Net.

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

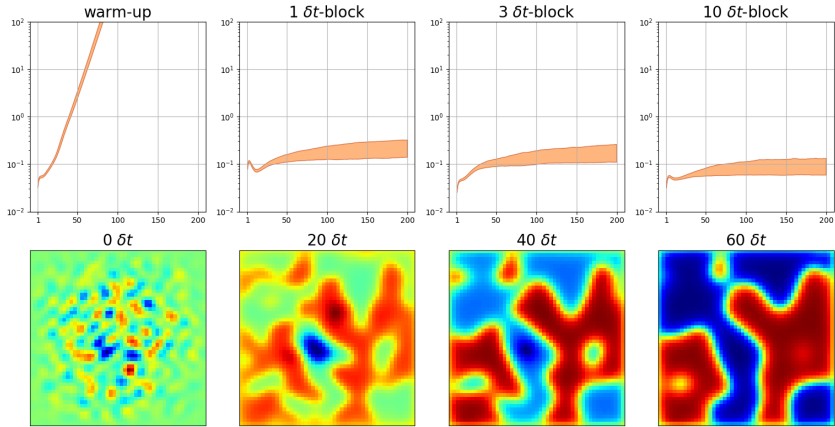

Figure 14: Testing with higher frequency initializations (diffusion equation with a nonlinear source). First row: long-time prediction. Second row: estimated dynamics.Here, $\delta t = 0.01$.

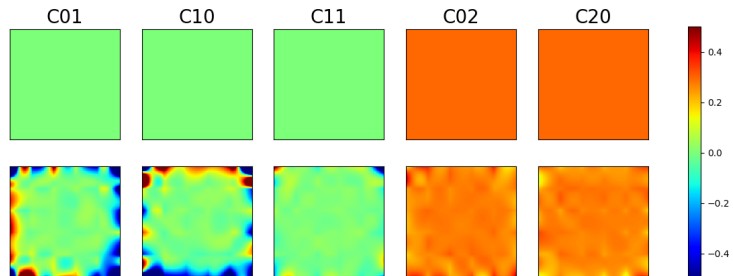

Figure 15: First row: the true coefficients $\{f_{ij} : 1 \leq i + j \leq 2\}$ of the equation. Second row: the learned coefficients $\{c_{ij} : 1 \leq i + j \leq 2\}$ by the PDE-Net with 3 $\delta t$-blocks and $7 \times 7$ filters.

*Sciences*, 113(15):3932–3937, 2016.

Jian-Feng Cai, Bin Dong, Stanley Osher, and Zuowei Shen. Image restoration: total variation, wavelet frames, and beyond. *Journal of the American Mathematical Society*, 25(4):1033–1089, 2012.

Subrahmanyan Chandrasekhar. Stochastic problems in physics and astronomy. *Reviews of modern physics*, 15(1):1, 1943.

Yunjin Chen, Wei Yu, and Thomas Pock. On learning optimized reaction diffusion processes for effective image restoration. In *Proceedings of the IEEE Conference on Computer Vision and Pattern Recognition*, pp. 5261–5269, 2015.

Ingrid Daubechies. *Ten lectures on wavelets*. SIAM, 1992.

Bin Dong, Qingtang Jiang, and Zuowei Shen. Image restoration: wavelet frame shrinkage, nonlinear evolution pdes, and beyond. *Multiscale Modeling & Simulation*, 15(1):606–660, 2017.

Weinan E. A proposal on machine learning via dynamical systems. *Communications in Mathematics and Statistics*, 5(1):1–11, 2017.

Ian Goodfellow, Yoshua Bengio, and Aaron Courville. *Deep learning*. MIT press, 2016.

Eldad Haber and Lars Ruthotto. Stable architectures for deep neural networks. *arXiv preprint arXiv:1705.03341*, 2017.

Kaiming He, Xiangyu Zhang, Shaoqing Ren, and Jian Sun. Deep residual learning for image recognition. In *Proceedings of the IEEE conference on computer vision and pattern recognition*, pp. 770–778, 2016a.

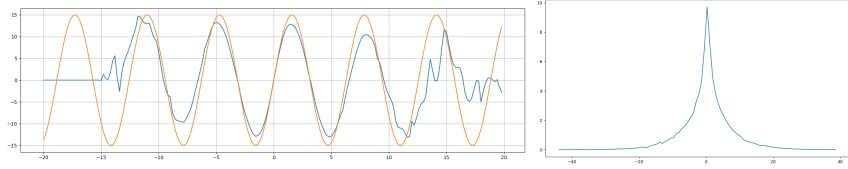

Figure 16: Left: the true source function $f_s$ and estimated source function $\tilde{f}_s$. Right: distribution of the values of $u$ during training.

Kaiming He, Xiangyu Zhang, Shaoqing Ren, and Jian Sun. Identity mappings in deep residual networks. In *European Conference on Computer Vision*, pp. 630–645. Springer, 2016b.

Min Lin, Qiang Chen, and Shuicheng Yan. Network in network. *arXiv preprint arXiv:1312.4400*, 2013.

Yiping Lu, Aoxiao Zhong, Quanzheng Li, and Bin Dong. Beyond finite layer neural networks: Bridging deep architectures and numerical differential equations. *ArXiv preprint*, 2017.

Stéphane Mallat. *A wavelet tour of signal processing*. Academic press, 1999.

Maziar Raissi and George Em Karniadakis. Hidden physics models: Machine learning of nonlinear partial differential equations. *arXiv preprint arXiv:1708.00588*, 2017.

Maziar Raissi, Paris Perdikaris, and George Em Karniadakis. Physics informed deep learning (part ii): Data-driven discovery of nonlinear partial differential equations. *arXiv preprint arXiv:1711.10566*, 2017.

Samuel H Rudy, Steven L Brunton, Joshua L Proctor, and J Nathan Kutz. Data-driven discovery of partial differential equations. *Science Advances*, 3(4):e1602614, 2017.

Hayden Schaeffer. Learning partial differential equations via data discovery and sparse optimization. In *Proc. R. Soc. A*, volume 473, pp. 20160446. The Royal Society, 2017.

Michael Schmidt and Hod Lipson. Distilling free-form natural laws from experimental data. *science*, 324(5923):81–85, 2009.

Sho Sonoda and Noboru Murata. Double continuum limit of deep neural networks. *ICML Workshop on Principled Approaches to Deep Learning, Sydney, Australia*, 2017.

Zongmin Wu and Ran Zhang. Learning physics by data for the motion of a sphere falling in a non-newtonian fluid non-newtonian fluid. *preprint*, 2017.

