# OpenReview forum: "PDE-Net: Learning PDEs from Data"
_ICLR.cc/2018/Conference — Invite to Workshop Track_

### Official Review · AnonReviewer3 · 2017-11-25
**Good paper on the use of deep learning for modeling and identifying dynamical PDE based systems**

**Rating:** 7
**Confidence:** 4

**Review:**

The paper explores the use of deep learning machinery for the purpose of identifying dynamical systems specified by PDEs.

The paper advocates the following approach:
One assumes a dynamic PDE system involving differential operators up to a given order. Each differential operator term is approximated by a filter whose values are properly constrained so as to correspond to finite difference approximations of the corresponding term. The paper discusses the underlying wavelet-related theory in detail. A certain form of the dynamic function and/or source terms is also assumed. An explicit Euler scheme is adopted for time discretization. The parameters of the system are learned by minimizing the approximation error at each timestep. In the experiments reported in the paper the reference signal is provided by numerical simulation of a ground truth system and the authors compare the prediction quality of different versions of their system (eg, for different kernel size).

Overall I find the paper good, well written and motivated. The advocated approach should be appealing for scientific applications of deep learning where not only the quality of approximation but also the interpretability of the identified model is important.

Some suggestions for improvement:
* The paper doesn't discuss the spatial boundary conditions. Please clarify this.
* The paper adopts a hybrid approach that lies in between the classic fully analytical PDE approach and the data driven machine learning. I would like to see a couple more experiments comparing the proposed approach with those extremes. (1) in the first experiment, the underlying model order is 2 but the experiment allows filters up to order 4. Can you please report if generalization quality improves if the correct order 2 is specified? (2) On the other side, what happens if no sum (vanishing order) constraints are enforced during model training? This abandons the interpretability of the model as approximating a PDE of given order but I am curious to see what is the generalization error of this less constrained system.

Nit: bibtex error in Weinan (2017) makes the paper appear as E (2017).

---

> ### Author Response · Authors · 2017-12-04
> **Response to the reviewer's comments**
>
> We would like to thank the reviewer for carefully evaluating the paper. The summary captures the main points of the paper. Our responses to the reviewers suggestions are as follows.
>
> 1, Boundary conditions are indeed very important for PDEs. In our PDE-Net, we did not emphasize much on dealing with complicated boundary conditions. To some extent, we can think that PDE-Net focuses on initial-value problems. However, for some simple initial-boundary problems, we can still apply PDE-Net by using padding strategies to deal with boundary conditions. For instance, in the first PDE example, we adopted the periodic boundary conditions and used periodic padding for the convolutions in the PDE-Net; in the second PDE example, we adopted the Dirichlet boundary condition and used zero padding.
>
> 2, We are working on the requested numerical experiments. We will upload a revised manuscript once they are done.
>
> 3, Weinan is the first name of the author. His last name is E and therefore the paper is cited as E (2017).

---

### Official Review · AnonReviewer1 · 2017-11-27
**Novel method, interesting approach, interpretability might be over-emphasized.**

**Rating:** 8
**Confidence:** 4

**Review:**

Authors propose a neural network based algorithm for learning from data that arises from dynamical systems with governing equations that can be written as partial differential equations.  The network architecture is constrained such that regardless of the parameters, it always implements discretization of an arbitrary PDE.  Through learning, the network adapts itself to solve a specific PDE.  Discretization is finite difference in space and forward Euler in time.

The article is quite novel in my opinion.  To the best of my knowledge, it is the first article that implements a generic method for learning arbitrary PDE models from data.  In using networks, the method differs from previously proposed approaches for learning PDEs.  Experiments are only presented with synthetic data but given the potential for the method and its novelty, I believe this can be accepted.  However, it would have been a stronger article if authors have applied to a real life model with real initial and boundary conditions, and real observations.

I have three main criticism about the article:

 1.   Authors do not cite Chen and Pock’s article on learning diffusion filters with networks that first published in CVPR 2015 and then authors published a PAMI article this year.  To the best of my knowledge, they are the first to show the connection between res-net type architecture and numerical solutions of PDEs. I think proper credit should be given. [I need to emphasize that I am not an author in that article.]
 2.   Authors emphasize the importance of interpretability, however, the constraint on the moment matrices might cripple this aspect.  The frozen filters have clear interpretations. They are in the end finite difference approximations with some level of accuracy depending on the size and the number of zeros.  When M(q) matrix is free to change, it is unclear what the effect will be on the filters.  Are the numbers that replace stars in Equation 6 for instance, will be absorbed in the O(\epsilon) term?  Can one really interpret the final c_{ij} for filters whose M(q) have many non-zeros?
 3.   The introduction and results sections are well written.  The method section on the other hand, needs improvement.  The notation is not easy to follow due to missing definitions.  I believe with proper definitions — amounting to small modifications — readability of the article can substantially improve.


In addition to the main criticisms, I have some other questions and concerns:

  1.  How sensitive is the model?  In real life, one cannot expect to get observations every delta t.  Data is most often very sparse.  Can the model learn in that regime?  Can it differentiate between different PDEs and find the correct one with sparse data?
  2.  The average operations decreases the interpretability of the proposed model as a PDE.  Depending on the filter size, D_{0}u can deviate from u, which should be the term that should be used in the residual block.  Why do authors need this?  How does the model behave without it?
  3.  The statement “Thus, the PDE-Net with bigger n owns a longer time stability.” is a very vague statement.  I understand with larger n, training would be easier since more data would be used to estimate parameters.  However, it is not clear how this relates to “time stability”, which is also not defined in the article.
 4.  How is the relative error computed?  Values in relative error plots goes as high as 10^2.  That would be a huge error if it is relative.

---

> ### Author Response · Authors · 2017-12-04
> **Response to the reviewer's comments**
>
> We would like to thank the reviewer for his/her constructive suggestions. Our responses are as follows.
>
> 1, Thanks for pointing this out! We are aware of Chen and Pock’s CVPR article. We did cite their work since they used numerical PDE (discretization of Perona-Malik) to inspire network architecture which is more related to the idea of unrolling dynamics originally proposed by Gregor and LeCun, ICML 2010. They did not require the underlying denoising process be governed by a PDE, nor did they attempt to recover the PDE (should there exist one). However, we do agree on the importance of Chen and Pock’s contribution, and will properly cite the paper in the revised version.
>
> 2, In PDE-Net, we only partly free M(q). On one hand, we impose zero-moment constrains on lower order moments so that we know which differential operator the corresponding convolution is approximating. On the other hand, we free higher order moments so that the filters can adjust themselves to achieve better stability and approximation accuracy according to the data. In this way, we are able to preserve some expressive power of setting all moments free (having full degree of freedom), while still maintain transparency of the network (i.e. knowing which filter is in correspondence to which differential operator so that we can identify the response function correctly). We have done experiments using the diffusion equation with a nonlinear source without any moment constraints. We got great prediction, whereas we cannot identify the equation at all.
>
> 3, Thanks for the suggestion! In the revision, we will improve clarity of the method section by including more definitions and explanations.
>
> For the reviewer’s other questions and concerns:
>
> 1, Since it’s hard for us to find suitable real 2D physical data, we have to test the idea on simulated data sets as a proof of concept. For sparse observations in time, if there are many replicated experiments with different initial values, we believe the PDE-Net is still effective though the depth may be limited by the number of temporal observations. If there is only one single experiment with few temporal observations, PDE-Net may fail due to lack of data. At this point, we cannot predict how much less data the PDE-Net can tolerate. But what we know for sure is we will need much less data than normal deep learning regime since PDE-Net has relatively fewer trainable parameters than heavy-duty networks in deep learning.
>
> 2, The idea of using D_0 u instead u in PDE-Net comes from stability of numerical PDEs. For a difference scheme of a PDE, $u_m^{n+1}=u_m^n+L_h(u)$, sometimes we use $1/2(u_{m+1}^n+u_{m-1}^n)$ instead of $u_m^n$ to get a modified scheme, which usually has a larger stable region, or it can even make an unstable scheme stable. Inspired by this, we introduce the average operator in PDE-Net. However, this treatment may or may not be vital depending on the data, but it has the potential to boost stability whenever it is needed.
>
> 3, When we apply the \delta t-block to a given data, the output has an error. The errors will accumulate when we repeatedly apply the \delta t-block. In general the error grows exponentially as shown by the blue error curves in Figure 3. By long time stability of a learned network (more precisely, the learned \delta t-block), we mean that the errors are well controlled after multiple \delta t blocks are applied. This is not exactly the same “stability” as in numerical PDEs, but it shares some similarities. To enable a longer time prediction, we demonstrated that if we train an n layer PDE-Net, rather than merely one or just a few \delta t-blocks, the filters will be learned to at least ensure the stability of applying \delta t-block n times. Choosing bigger n for the PDE-Net indeed helps to slow down the growth of the prediction error, which was demonstrated in for instance Figure 3&8.
>
> 4, The relative error in our paper is defined by $\epsilon_r=\frac{\sum(\tilde{u}_m-u_m)^2}{\sum(u_m-\bar{u})^2}$, where u is the true data, \bar{u} is the average of u, and \tilde{u} is the predicted data. Maybe calling it “normalized error” is less misleading, and we will correct it in the revised version.
>
> P.S. we are working on the revision. Once we are done with the requested additional experiments, we will summarize the new results along with other suggested modifications in the revised manuscript.

---

### Official Review · AnonReviewer2 · 2017-11-30
**A promising approach on nonparametric modelling of partial differential equations with deep architectures that requires more details.**

**Rating:** 5
**Confidence:** 4

**Review:**

This paper addresses complex dynamical systems modelling through nonparametric Partial Differential Equations using neural architectures. It falls down within the context of a recent and growing literature on the subject.

The most important idea of the papier (PDE-net) is to learn both differential operators and the function that governs the PDE. To achieve this goal, the approach relies on the approximation of differential operators by convolution of filters of appropriate order.  This is really the strongest point of the paper.

Moreover, a basic system called delta t block implements one level of full approximation and is stoked several times.
A short section relates this work to recent existing work and numerical results are deployed on simulated data.
In particular, the interest of learning the filters involved in the approximation of the differential operators is tested against a frozen variant of the PDE-net.

Comments:
The paper is badly structured and is sometimes hard to read because it does not present in a linear way the classic ingredients of Machine Learning, expression of the full function to be estimated, equations of each layer, description of the set of parameters to be learned and the loss function. Parts of the puzzle have to be found in the core of the paper as well as in simulations.

About the loss function, I was surprised not to see a sparsity constraint on the different filters in order to select the order of the differential operators themselves. If one want to achieve interpretability of the resulting PDE, this is very important.

I also found difficult to measure the degree of novelty of the approach considering the recent works and  the related work section should have been much more precise in terms of comparison.

For  the simulations, it is perfectly fine to rely on simulated datasets. However the approach is not compared to the closest works (Sonoda et al., for instance).

Finally, I’ve found the paper very interesting and promising but regarding the standard of scientific publication, it requires additional attention to provide a better description the model and discuss the learning scheme to get a strongest and reproducible approach.

---

> ### Author Response · Authors · 2017-12-04
> **Responses to reviewer's original comments**
>
> First, we would like to thank the reviewer for carefully evaluating our paper. The reviewer’s summary captures most points of our paper. However, we think there are still some details and innovations of the paper that are not noticed, and we will respond to the reviewer’s comments in detail.
>
> 1, We understand that different community organizes papers in rather different ways. The subject we study crosses the field of machine learning and applied mathematics. Thus, we had to take the conventions of both fields into account when organizing the manuscript. In the current form of the paper, we first clearly described the PDE to be estimated (Eq.(1)). Then we link convolution with differential operators and state the necessary notions needed before we can introduce our entire network (in Section 2.1). Since the notions are the key to grant transparency to the PDE-Net while preserving its expressive power, it deserves a separate subsection. Then we introduced the network architecture and the loss function in Section 2.2, followed by the discussions on parameters and initialization in Section 2.3. In order to illuminate our thoughts more clearly, we had to state the framework in a general setting first, and show detailed and intricate implementations for each special numerical experiment.
>
> 2, Sparsity is really important for selecting the simplest form from dictionaries, just as in symbolic regression (Bongard & Lipson,2007) and sparse regression (Brunton et al. ,2016). For the neural networks, sparsity is also a popular choice of regularization. However, for PDE-Net, our numerical experiments show that it already performs well and has a good generalization even without a sparsity based regularization. We also note that the total number of trainable parameters is smaller than most deep networks. Therefore, we do not need sparsity to further reduce the space of parameters to prevent overfitting.
>
> 3, As clearly stated in the introduction, the existing work on learning PDEs from data requires either a fixed (non-trainable) numerical approximation of derivatives (Rudy et al. ,2017), or knowing the exact form of the nonlinear response function (Raissi &Karniadakis , 2017). For PDE-Net, however, unlike the existing work, the proposed network only requires minor knowledge on the form of the nonlinear response function, and requires no knowledge on the involved differential operators (except for their maximum possible order) and their associated discrete approximations. We think it is a breakthrough comparing to the existing work. To the best of our knowledge, it’s also the first time to introduce the relation between sum rule of filters and the order of differential operators into the design of neural networks.
>
> 4, As for the “relation to some existing networks”, we think they do not have much to do with learning PDEs from data, since most of those work do not assume (or it’s simply untrue) that the underlying process is governed by some PDE. We have also noticed Sonoda’s contributions in this area. They interpreted continuous denoising autoencoder as an approximation to backward heat equation applied to the distribution of the data set (Sonoda& Murata, 2016). Overall, these networks and their analysis focus on rather different aspects from our task.
>
> 5, Learning PDEs from data is a rather challenging task if you want both predictive power and transparency. Therefore, it is worthwhile to first investigate the viability of the proposed approach on simulated data sets so that we can have precise evaluations of the performance. Furthermore, it’s very hard for us to find suitable real 2D physical data, we have to test the idea first on simulated data sets. Application to real-world datasets is definitely one of our future directions.
>
> At last, thanks again for the reviewer’s comments and suggestions. But we still strongly believe that this manuscript is meaningful and deserves publishing.

---

### Author Response · Authors · 2018-01-01
**Manuscript updated**

Dear Area Chair and Reviewers,

We have revised our manuscript according to the reviewers suggestions. In particular, we have

1) added description to some of the notation. In particular, we added some examples after Proposition 2.1 to help the readers understand the concept of sum rules and its relation to differential operators.

2) added some experiments in Section 3 (starting from page 11, "Further Experiments"). We compared the original PDE-Net with PDE-Net assuming we know the highest order of the linear PDE is 2. We also compared the PDE-Net with the Freed-PDE-Net (the network without any moment constraints). In a nutshell, with more prior knowledge on the unknown PDE, we are able to obtain a more accurate estimation on the model. Also, having no moment constraints, we cannot identify the PDE model, though we are be able to improve the prediction accuracy over the original PDE-Net.

---

### Decision · Program_Chairs · 2018-01-29
**ICLR 2018 Conference Acceptance Decision**

**Decision:**

Invite to Workshop Track

**Comment:**

This paper studies the approximation and integration of partial differential equations using convolutional neural networks. By constraining CNN filters to have prescribed vanishing moments, the authors interpret CNN-based temporal prediction in terms of 'pde discovery'. The method is demonstrated on simple convection-diffusion simulations.

Reviewers were mixed in assessing the quality, novelty and significance of this work. While they all acknowledged the importance of future research in this area, they raised concerns about clarity of exposition (which has been improved during the rebuttal period), as well as the novelty / motivation. The AC shares these concerns; in particular, he misses a more thorough analysis of stability (under what conditions would one use this method to estimate an actual PDE and obtain some certificate of approximation?) and discussions about pitfalls (in real situations one may not know in advance the family of differential operators involved in the physical process nor the nature of the non-linearity; does the method produce a faithful approximation? why?).

Overall, the AC thinks this is an interesting submission that is still in its preliminary stage, and therefore recommends resubmitting to the worshop track at this time.